# SOX2 Expression Is an Independent Predictor of Oral Cancer Progression

**DOI:** 10.3390/jcm8101744

**Published:** 2019-10-21

**Authors:** Juan C. de Vicente, Paula Donate-Pérez del Molino, Juan P. Rodrigo, Eva Allonca, Francisco Hermida-Prado, Rocío Granda-Díaz, Tania Rodríguez Santamarta, Juana M. García-Pedrero

**Affiliations:** 1Department of Oral and Maxillofacial Surgery, Hospital Universitario Central de Asturias (HUCA). C/Carretera de Rubín, s/n, 33011 Oviedo, Asturias, Spain; pauladonatepdm@gmail.com (P.D.-P.d.M.); taniasantamarta@gmail.com (T.R.S.); 2Department of Surgery, University of Oviedo. Avda. Julián Clavería, s/n, 33006 Oviedo, Asturias, Spain; jprodrigot@telefonica.net; 3Instituto de Investigación Sanitaria del Principado de Asturias (ISPA), Instituto Universitario de Oncología del Principado de Asturias (IUOPA), Universidad de Oviedo. C/Carretera de Rubín, s/n, 33011 Oviedo, Asturias, Spain; ynkc1@hotmail.com (E.A.); franjhermida@gmail.com (F.H.-P.); rocigd281@gmail.com (R.G.-D.); 4Department of Otolaryngology, Hospital Universitario Central de Asturias (HUCA). C/Carretera de Rubín, s/n, 33011 Oviedo, Asturias, Spain; 5Ciber de Cáncer (CIBERONC), Instituto de Salud Carlos III, Av. Monforte de Lemos, 3-5. 28029 Madrid, Spain

**Keywords:** oral cancer risk, oral epithelial dysplasia, SOX2, immunohistochemistry, oral squamous cell carcinoma

## Abstract

Potentially malignant oral lesions, mainly leukoplakia, are common. Malignant transformation varies widely, even in the absence of histological features such as dysplasia. Hence, there is a need for novel biomarker-based systems to more accurately predict the risk of cancer progression. The pluripotency transcription factor SOX2 is frequently overexpressed in cancers, including oral squamous cell carcinoma (OSCC), thereby providing a link between malignancy and stemness. This study investigates the clinical relevance of SOX2 protein expression in early stages of oral carcinogenesis as a cancer risk biomarker, and also its impact on prognosis and disease outcome at late stages of OSCC progression. SOX2 expression was evaluated by immunohistochemistry in 55 patients with oral epithelial dysplasia, and in 125 patients with OSCC, and correlated with clinicopathological data and outcomes. Nuclear SOX2 expression was detected in four (7%) cases of oral epithelial dysplasia, using a cut-off of 10% stained nuclei, and in 16 (29%) cases when any positive nuclei was evaluated. Univariate analysis showed that SOX2 expression and histopathological grading were significantly associated with oral cancer risk; and both were found to be significant independent predictors in the multivariate analysis. Nuclear SOX2 expression was also found in 49 (39%) OSCC cases, was more frequent in early tumor stages and N0 cases, and was associated with a better survival. In conclusion, SOX2 expression emerges as an independent predictor of oral cancer risk in patients with oral leukoplakia. These findings underscore the relevant role of SOX2 in early oral tumorigenesis rather than in tumor progression.

## 1. Introduction

Squamous cell carcinoma (SCC) of the oral cavity (OSCC) afflicts about 300,400 new cases and causes 145,400 deaths worldwide each year [1], with a predilection for South Asian and Southeast Asian populations [2]. According to the World Health Organization (WHO) mortality projections, there is an estimate of 679,941 mouth and oropharynx cancer-related deaths by 2060 [3]. OSCC shows an aggressive growth pattern with a high degree of local invasiveness and a propensity to metastasize to the cervical lymph nodes, even in early stages. In fact, metastases to neck lymph nodes occur in 40% of cases, which remain the main factor associated with poor prognosis [4]. Additionally, between 26% and 80% of patients with early-stage OSCC develop locoregional recurrence or distant metastasis [4]. Despite aggressive treatment, the prognosis of this disease remains dismal, with a five-year survival rate at around 55%–60% [2]. 

OSCC may develop from an apparently normal oral mucosa or from oral potentially malignant disorders (OPMDs), mainly oral leukoplakia (OLK), which shows a malignant transformation rate of 0.1% to 36% [5]. All premalignant lesions, such as OLK, erythroplakia, submucosal fibrosis, or oral lichen planus, may harbor many genetic alterations present in OSCC [6].

Oral cancer exhibits cellular heterogeneity and is composed by three different types of cells including highly differentiated bulk tumor cells, transit cells with maximum proliferation capacity, and a small subpopulation of cells with elevated self-renewal capacity and plasticity called cancer stem cells (CSCs) [7]. The CSCs are capable of long-term self-renewal and generation of the phenotypically diverse tumor cell population [8], and may be responsible for the genesis, anchorage-independent growth, cellular migration, and metastatic spread of the tumor [9,10]. Meantime, investigations by Takahashi and Yamanaka [11] revealed that the expression of four transcription factors (SOX2, Oct4, c-Myc, and Klf4) was sufficient to reprogram differentiated cells into induced pluripotent stem cells (iPSCs). One of these genes, the so-called sex-determining region Y (SRY)-related high-mobility-group (HMG)-box 2 (SOX2), located on chromosome 3q26, is implicated in the maintenance of embryonic stem cell pluripotency [12]. Numerous evidences indicate that SOX2 is involved in tumorigenesis, thereby acting as a link between malignancy and stemness [13]. Moreover, the proliferation of CSCs in head and neck SCC was inhibited both in vitro and in vivo when SOX2 was suppressed by all-trans-retinoic acid [14]. The degree of similarity to OSCC found in premalignant lesions depend upon the presence of atypia; however, individual lesions exhibit molecular genetic traits in common with OSCC, even in the absence of histologically-defined dysplasia [15]. Hence, better biomarker-based detection systems need to be developed to more accurately predict the risk of cancer progression in potentially malignant oral disorders. 

It has been described that OLK lesions show higher expression of SOX2 than normal oral mucosa, suggesting its contribution to the pathogenesis of OSCC [16]. Consequently, SOX2 could be a potentially useful predictor of cancer risk in the oral cavity. SOX2 is mainly expressed in CSCs [13], and is one of the amplified genes in OSCC, where its expression has been closely associated with lymph node metastasis [17] and poor prognosis [18]. In marked contrast, several studies have demonstrated that increased levels of SOX2 were significantly associated with better prognosis in patients with OSCC, and also in squamous cell lung cancer [19,20]. Therefore, the role of SOX2 expression in OSCC prognosis remains controversial. 

This prompted us to perform a thorough study to investigate the clinical significance of SOX2 in the development and progression of OSCC. To accomplish this, the expression pattern of SOX2 was evaluated at different stages of oral tumorigenesis, from potentially malignant oral disorders (i.e., oral epithelial dysplastic lesions) to invasive carcinomas, to ascertain its contribution to tumor initiation and malignant transformation, and also late stages of disease progression. 

## 2. Materials and Methods

### 2.1. Patients and Tissue Specimens

Surgical tissue specimens from 55 patients who were diagnosed with oral mucosa dysplasia at the Hospital Universitario Central de Asturias between 2000 and 2005 were retrospectively collected. All selected patients met the following inclusion criteria: (i) pathological diagnosis of oral epithelial dysplasia; (ii) lesions of the oral mucosa (leukoplakia); (iii) no previous history of head and neck cancer, (iv) complete excisional biopsy of the lesion; and (v) a minimum follow-up of five years (or until progression). Patients were followed up as previously described [21]. 

In addition, surgical tissue specimens from 125 patients with histologically-confirmed OSCC surgically treated at the Hospital Universitario Central de Asturias between 1996 and 2007 were retrospectively collected, in accordance to approved institutional review board guidelines. All experimental procedures were conducted in accordance to the Declaration of Helsinki and approved by the Institutional Ethics Committee of the Hospital Universitario Central de Asturias and by the Regional (CEIC) from Principado de Asturias (date of approval 5th of May 2016; approval number: 70/16) for the project PI16/00280. Informed consent was obtained from all patients. 

Tissue samples and data from donors were provided by the Principado de Asturias BioBank (PT17/0015/0023), integrated in the Spanish National Biobanks Network, and processed following standard operating procedures with the appropriate approval of the Ethical and Scientific Committees. Representative tissue samples were obtained from archival, formalin-fixed, paraffin-embedded blocks and the histological diagnosis was confirmed by an experienced pathologist. 

### 2.2. Tissue Microarray (TMA) Construction

As previously described [21], three representative tissue cores (1 mm diameter) were taken from each tumor block to construct OSCC TMAs. Each TMA block also included three cores of normal epithelium as an internal control. These samples were obtained from non-oncological patients undergoing oral surgery. 

### 2.3. Immunohistochemistry (IHC)

The TMAs were cut into 3 μm sections and dried on Flex IHC microscope slides (DakoCytomation, Glostrup, Denmark). Antigen retrieval was performed by heating the sections with Envision Flex Target Retrieval solution, high pH (Dako, Glostrup, Denmark). Staining was done at room temperature on an automatic staining workstation (Dako Autostainer Plus, Glostrup, Denmark) with anti-SOX2 rabbit polyclonal antibody (AB5603, Merck Millipore, Darmstadt, Germany) at 1:1000 dilution using the Dako EnVision Flex + Visualization System (Dako Autostainer, Glostrup, Denmark) and diaminobenzidine chromogen as substrate. Counterstaining with hematoxylin was the final step. 

The IHC results were independently evaluated by two observers (JPR, and JMG-P), blinded to clinical data. SOX2 staining was evaluated as the percentage of cells with nuclear staining in the dysplastic epithelium or in the tumor tissue. SOX2 staining scores were classified as negative or positive staining on the basis of values below or above the median value of 10%. Since CSC-like subpopulations are usually limited to a very small percentage of cells, SOX2 staining in the dysplastic areas was also scored considering any positive nuclei.

### 2.4. Statistical Analysis

Bivariate analysis by χ^2^ and Fisher’s exact tests were used for comparison between SOX2 expression and clinicopathological categorical variables. Disease-specific survival (DSS) was determined from the date of treatment completion to the death of the tumor. For time-to-event analysis, survival curves were estimated using the Kaplan–Meier method. The log-rank test was used to compare the survival curves. Hazard ratios (HRs) with their 95% confidence intervals (CIs) for clinicopathological variables were calculated using univariate and multivariate Cox proportional hazards model. All tests were two-sided and *P*-values less than 0.05 were considered statistically significant. All statistical analyses were performed using SPSS version 18 (IBM Co., Armonk, NY, USA).

## 3. Results

### 3.1. Clinicopathological Features and Follow-Up in Patients with Oral Epithelial Dysplasia 

Twenty-six patients (47%) were men and the remaining 29 were women (53%), with a mean age of 62.61 years (SD 12.56, range 39 to 83 years). Ten patients (18%) were smokers and four (7%) were habitual alcohol drinkers. Forty-two of 55 premalignant lesions (76%) were classified as mild dysplasia, six (11%) as moderate dysplasia, and the remaining seven (13%) as severe dysplasia, according to the WHO classification [22]. During the follow-up period (mean: 85.47, SD: 44.41, median: 75, range: four to 252 months), 12 (22%) of 55 patients developed an invasive OSCC. The most relevant clinical and pathological characteristics are summarized in Appendix A.

### 3.2. SOX2 Protein Expression in Oral Epithelial Dysplasia

Nuclear SOX2 expression was detected in four (7%) cases when a cut-off of 10% stained nuclei was used (SOX2 > 10), and in 16 (29%) when any positive nuclei was applied (SOX2any). Normal adjacent epithelia showed negative SOX2 expression (Figure 1A–C). SOX2 protein expression was found to significantly increase with the grade of dysplasia (Table 1). In addition to the WHO three-tier classification, the binary WHO grading system (low-grade vs. high-grade) was used and correlated with SOX2 expression.

There was a statistically significant correlation between the histopathological grade (both the WHO histological classification and the binary dysplasia grading) and the risk of progression to oral cancer in this cohort (log-rank test, *P* < 0.001; Figure 2A,B) (Table 2). In addition, positive SOX2 expression also significantly predicted oral cancer risk either considering SOX2 > 10 (log-rank test, *P* = 0.02; Figure 2C) or SOX2any (log-rank test, *P* = 0.01; Figure 2D) as cut-off points. Univariate Kaplan–Meier and Cox analysis showed that the SOX2 expression and histological grading were significantly associated with oral cancer risk (Table 3). When these factors were simultaneously analyzed using a multivariate Cox analysis, only SOX2 expression calculated using SOX2any as the cut-off point and the dysplasia grading were significant independent predictors of OSCC development (Table 4).

Similarly, we have recently reported a novel role for another pluripotency factor NANOG as a cancer risk marker using the same subset of 55 patients with oral epithelial dysplasia [21]. Since SOX2 and NANOG are functionally-related proteins, this prompted us to assess the impact of combined expression of SOX2 and NANOG in regards to malignization risk. Interestingly, results consistently showed that patients harboring positive expression of SOX2 (either SOX2 > 10% or SOX2any) and NANOG (either cytoplasmic or nuclear expression) significantly exhibited a much higher risk of developing oral cancer, compared to patients with positive expression of either SOX2 or NANOG or those patients with negative expression (Table 5). 

### 3.3. Clinicopathological Features and Follow-Up in the Cohort of OSCC Patients

The mean and median follow-up times were 71.82 (SD: 57.55) and 61.0, respectively. Neck node metastases were present in 49 (39%) cases, and local recurrences were found in 54 (43%) cases. No patient had distant metastasis at the time of diagnosis. The five- and 10-year disease-specific survival rates were 60% and 44%, respectively. The mean and median survival times were 132.74 months (95% CI: 113.25 to 152.22 months) and 141 months (95% CI: 102.40 to 179.59 months), respectively. The remaining relevant clinical and pathological characteristics are summarized in Appendix A.

### 3.4. SOX2 Protein Expression and Its Relation with Clinicopathological Variables and Follow-Up

Positive SOX2 staining was found in 49 (39%) cases located in the nucleus of tumor cells, whereas stromal cells and normal epithelium showed negligible expression (Figure 1A,D,E). SOX2 expression did not show any significant association with the clinicopathological variables studied (Appendix A). In the survival analysis, tumor size and local extension (pT), neck node status (pN), and stage were significantly correlated to survival (Appendix A). Positive SOX2 expression was associated with a longer disease-specific survival, although differences did not reach statistical significance (log-rank test, *P* = 0.07; Figure 3). 

### 3.5. In Silico Analysis of SOX2 mRNA Expression and Copy Number Alterations using the Cancer Genome Atlas (TCGA) Data

The role of SOX2 mRNA expression and copy number alterations in OSCC was investigated by analyzing a subset of 172 OSCC patients from the TCGA Head and Neck Squamous Cell Carcinoma (HNSCC) cohort [23] using the platform cBioPortal (http://cbioportal.org/) [24]. As shown in Figure 4A, SOX2 gene alterations were present in a total of 38 (22%) of 172 OSCC patients. Twenty-two (13%) cases harbored SOX2 mRNA up-regulation as previously reported [21], and 23 (13%) cases harbored *SOX2* gene amplification. SOX2 mRNA levels were associated to copy number alterations (Figure 4B). Overall, SOX2-amplified tumors showed higher mRNA levels; however, SOX2 amplification was only concomitantly accompanied by gene expression up-regulation in eight cases (5%). The impact of SOX2 mRNA expression on OSCC patient survival was also assessed (Figure 4C). The median survival times for patients with high (above the median) and low SOX2 mRNA levels (below the median) were 26.41 and 19.19 months, respectively, although differences did not reach statistical significance (*P* = 0.495, log-rank test). 

## 4. Discussion

To the best of our knowledge this is the first study to investigate SOX2 protein expression along the different stages of oral carcinogenesis, from potentially malignant oral disorders, such as leukoplakia, to invasive carcinomas, to ascertain its contribution to tumor initiation and malignant transformation, and also late stages of disease progression. 

Cancer stem cells (CSCs) are defined as a small subpopulation of cells in the tumors that possess the ability to initiate neoplasms and sustain tumor self-renewal [7]. *SOX2* gene mapping at 3q26 is frequently amplified in OSCC and other cancers. It has been established as an important CSC marker and a key molecule in the development of tumorigenesis in various cancers [13] and thus proposed as an oncogene [25,26]. Arnold et al. [27] reported that epithelial adult stem cells expressing SOX2 may be residual stem niches that originate from embryonic SOX2-positive tissue progenitors. Cai et al. [28] investigated the roles of OCT4 and SOX2 in the reprogramming of oral cancer stem cells. They immortalized oral epithelial cells by lentiviral transduction and found that double-transduced OCT4^+^SOX2^+^ cells were able to trigger tumor formation in immunodeficient mice; however, single-transduced OCT4^+^ or SOX2^+^ cells did not show tumorigenic capacity. They also stated that oral carcinogenesis may derive from OCT4^+^SOX2^+^ reprogrammed stem cells, in which SOX2 plays a major role in the regulation of the CSC niche [28]. Accordingly, it has been proposed that, in the absence of SOX2 expression, CSC self-renewal that sustains tumor growth could be abrogated,; therefore, supporting SOX2 inhibition as a potentially relevant therapeutic target for oral cancer [28]. 

OLK is the most frequent potentially malignant disorder in the oral cavity. Histologic grading of epithelial dysplasia in OLK is currently still the gold standard in the clinical practice to evaluate the risk of progression to invasive carcinoma [16]. However, the accuracy of the various grading systems so far developed have shown limited predictability, are largely subjective, and affected by a great inter- and intra-examiner variability [29]. According to the WHO 2017 classification there is not a unique criteria or grading system for oral epithelial dysplasia, and as a consequence diagnostic reproducibility is still limited. In addition to the currently accepted WHO three-tier classification for oral epithelial dysplasia, a binary grading system (low-grade vs. high-grade) has also been proposed [22]. However, this binary system has not yet been validated for use in the oral cavity. It is; therefore, of paramount importance to identify novel biomarkers that could provide complementary information to histology to more accurately predict the risk of malignant transformation of OLK. SOX2 has been demonstrated to play a central role in the maintenance of CSC pluripotency and self-renewal [27], thereby emerging as a promising marker for oral carcinogenesis. Luiz et al. [16] conducted a retrospective study to compare the expression of SOX2 in OLK with normal oral mucosa, and found that SOX2 expression was higher in OLK, although the relationship with oral cancer risk was not evaluated. 

The presence of dysplastic features in the epithelium of the oral cavity are thought to be relevant to malignant transformation in OLK. In our study, patients harboring high-grade dysplasia indeed showed a significantly higher risk of malignant progression (HR = 19.08). SOX2 expression was also found to be a significant predictor of risk of cancer development (HRs of 6.13 and 5.75 depending on the cut-off used). Furthermore, dysplasia grading and SOX2 expression were both found to be significant independent predictors of oral cancer risk in multivariate analysis. Similarly, we recently demonstrated that NANOG expression was a novel cancer risk marker using the same subset of patients with oral epithelial dysplasia [21]. Importantly, this study further extends these data to show that patients harboring positive expression of both SOX2 and NANOG significantly exhibited a much higher risk of progression to oral cancer, thereby suggesting a cooperative oncogenic role of these two proteins in oral pathogenesis and malignant transformation.

SOX2 expression has also been analyzed during tumor progression. Freier et al. [30] reported SOX2 expression in 18% of OSCC, and other studies on SCC of larynx, pharynx, and oral cavity found frequencies up to 86% [22,31,32,33]. In our cohort of 125 OSCC cases, positive SOX2 expression was detected in 39% of cases, while the frequencies of SOX2 expression reported in a previous study, performed at our laboratory for other head and neck subsites, were 38% in hypopharynx, 42% in larynx, and 14% in sinonasal cancer. In consequence, SOX2 expression frequencies in the oral cavity were similar to those observed in neighboring tissues such as hypopharynx and larynx, but much higher than in the sinonasal tract. In silico analysis of RNAseq data in a subset of 172 OSCC patients from the TCGA HNSCC cohort [23] further contributed to demonstrate that SOX2 mRNA levels were up-regulated in 13% of OSCC patients. In addition, SOX2 gene amplification was also observed in 13% of cases; however, only 5% of cases harboring SOX2 amplification were concomitantly accompanied by increased gene expression, indicating that additional mechanisms must be contributing to SOX2 expression in OSCC. In this sense, there are various plausible transcriptional regulatory mechanisms, such as the transcription factors OCT4 and YAP1 or the hypoxic factor HIF1α, found to modulate SOX2 expression [34,35]. Moreover, SOX2 protein expression was detected at much higher percentages (38%) in our OSCC cohort than SOX2 mRNA expression (13%), thereby suggesting the possible involvement of post-transcriptional mechanisms. Similar observations have been reported for other CSC-related factors, such as NANOG or PDPN, detected in over 30% of OSCC patients at the protein level [21,36] compared to 3% at mRNA levels according to the TCGA data from 172 OSCC patients [21]. 

Studies published to date assessing the expression of SOX2 and its clinical and prognostic relevance in OSCC have led to contradictory results. It has been reported that a high expression of SOX2 was significantly associated with poorer survival in node-negative OSCC [18], while others [17,26,37] found that a high expression of SOX2 correlated with lymph node metastasis. The latter could be explained by the relevant role of SOX2 as an epithelial–mesenchymal transition (EMT) inducer, thereby acting through the Wnt/β-catenin and phosphoinositide 3-kinase/protein kinase B signaling pathways [38,39] to promote oncogenesis, invasion, and metastasis. Nonetheless, the opposite viewpoint is represented by patients with early-stage OSCC, where upregulation of SOX2 has been correlated with a lower incidence of lymph node metastasis [2,20]. These controversial results could be explained by the heterogeneity of different tumors, as well as by different molecular mechanisms underlying the complex process of metastasis. The majority of studies showed that SOX2 overexpression may promote cancer progression; however, it has also been reported that SOX2 overexpression inhibits cell proliferation [13]. Furthermore, SOX2 is thought to stabilize stem cell phenotype and prevent EMT [40], and even more, SOX2 could promote mesenchymal–epithelial transition (MET), attenuating the invasive phenotype [41]. In our study, the expression of SOX2 did not correlate to T classification, neck lymph node metastasis, disease stage, histological grade, tumor recurrence, and second primary carcinomas development. 

Most studies showed that the survival rate of OSCC patients harboring low levels of SOX2 increased compared to those with high levels of SOX2 [13], although Züllig et al. [20] and Fu et al. [2] found that low expression of SOX2 was significantly associated with worse survival. Similarly, other studies on SCC from various locations described an association of elevated SOX2 expression with longer survival [19,42,43,44]. We herein observed that SOX2 expression was more frequently observed in early tumor stages as well as in N0 cases, and patients with positive SOX2 expression exhibited a better survival than those with negative expression, although differences did not reach statistical significance (*P* = 0.07).

It is possible that all these contradictory results are related to methodological differences in SOX2 IHC scoring, as well as in the heterogeneity of patient populations or even in the morphological and genetic heterogeneity observed in solid tumors. Another explanation may be that SOX2 copy number gain and expression are early tumor-initiating events, but this gene can lose its relevance in conveying an aggressive or metastasizing phenotype [22].

## 5. Conclusions

Together, our results reveal that SOX2 expression is a clinically relevant feature in early stages of oral tumorigenesis, and provide original evidence of its potential utility as biomarker for oral cancer risk assessment. According to these data, SOX2 expression emerges as an important determinant in the pathogenesis of a subset of OSCC, thereby contributing to tumor initiation and acquisition of an invasive phenotype rather than late stages of disease progression.

## Figures and Tables

**Figure 1 jcm-08-01744-f001:**
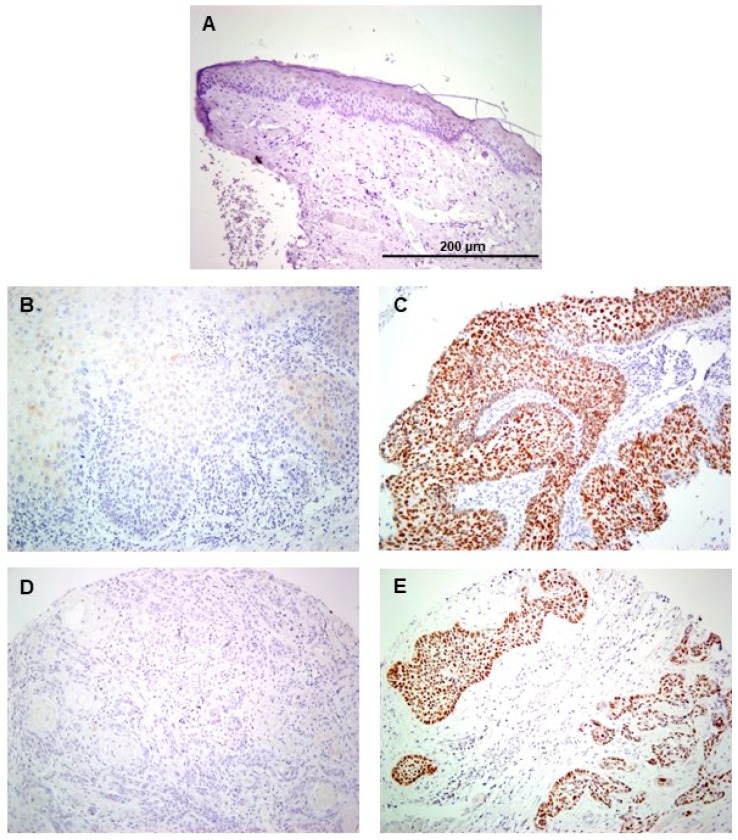
Immunohistochemical analysis of SOX2 expression in oral epithelial dysplasia. Normal adjacent epithelia showed negative staining (**A**). Representative examples of oral dysplasia showing negative (**B**) and positive nuclear SOX2 staining (**C**), and oral squamous cell carcinomas with negative (**D**) and positive SOX2 staining (**E**). Magnification 20 ×; scale bar 200 µm.

**Figure 2 jcm-08-01744-f002:**
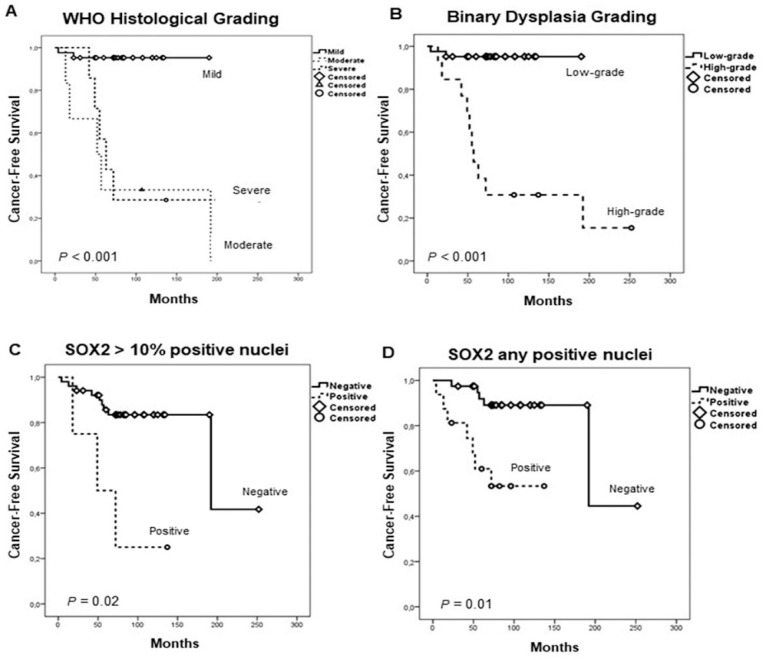
Kaplan–Meier cancer-free survival curves in the cohort of 55 patients with oral epithelial dysplasia categorized by the World Health Organization (WHO) histological grading (**A**), the binary dysplasia grading (**B**), and SOX2 protein expression dichotomized using the cut-off values of SOX2 staining > 10% positive nuclei (**C**) or SOX2 staining any positive nuclei (**D**).

**Figure 3 jcm-08-01744-f003:**
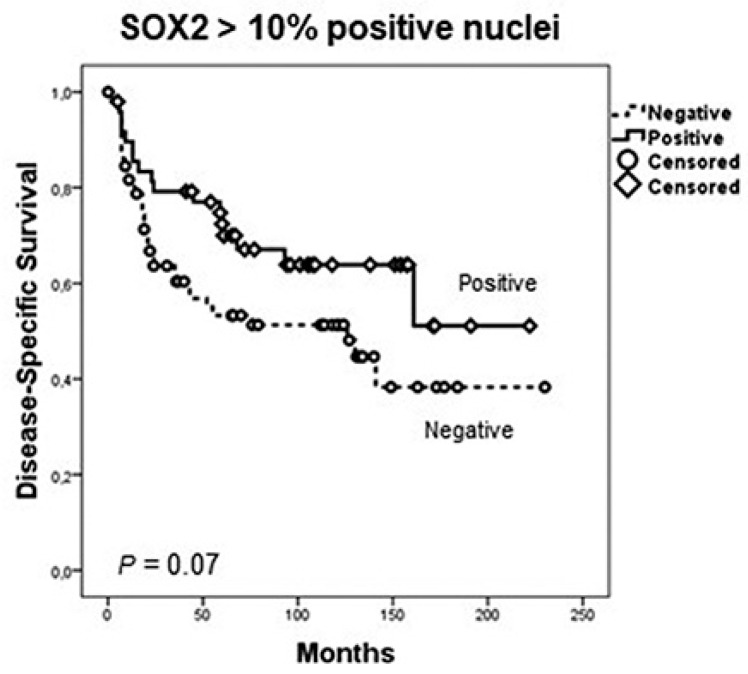
Kaplan–Meier disease-specific survival curves in the cohort of 125 patients with oral squamous cell carcinoma dichotomized according to SOX2 staining (positive versus negative). *P*-values were estimated using the log-rank test.

**Figure 4 jcm-08-01744-f004:**
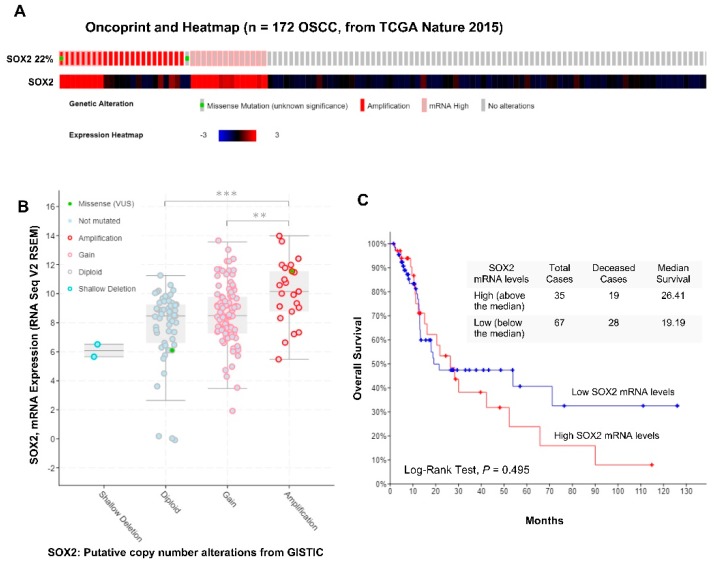
In silico analysis of mRNA expression and copy number alterations of SOX2 in the subset of 172 oral squamous cell carcinoma patients from The Cancer Genome Atlas (TCGA) Head and Neck Squamous Cell Carcinoma cohort [23] using the platform cBioPortal. (**A**) Oncoprint and heatmap representations showing the percentage of cases with SOX2 gene amplification, mutation, and mRNA up-regulation. (**B**) SOX2 mRNA expression analysis in relation to the copy number alterations of SOX2 gene (RNA seq V2 RSEM) values were Log2 transformed (*y*-axis). Whiskers plot (min. to max.) with median values; ** *P* < 0.01 and *** *P* < 0.001, one-way ANOVA, Tukey’s test. (**C**) Kaplan–Meier survival curves categorized by SOX2 mRNA expression (RNA seq V2 RSEM, z-score threshold ± 2) dichotomized as high mRNA levels (above the median) versus low mRNA levels (below the median), *P*-value estimated using the log-rank test.

**Table 1 jcm-08-01744-t001:** Associations between SOX2 expression and patient characteristics.

Characteristics	SOX2 > 10% Positive Nuclei Negative Positive		*P*	SOX2 any Positive Nuclei Negative Positive		*P*
Age (years), Mean (SD)	62.93 (12.69)	60.50 (13.20)	0.72	61.00 (12.69)	65.55 (12.35)	0.34
Gender, number (%)						
• Female	27 (93)	2 (7)	1.00	22 (76)	7 (24)	0.39
• Male	24 (92)	2 (8)		17 (65)	9 (35)	
Smoking, number (%)						
• Yes	9 (90)	1 (10)	1.00	6 (60)	4 (40)	0.71
• No	18 (86)	3 (14)		14 (67)	7 (33)	
Ethanol intake, number (%)						
• Yes	3 (75)	1 (25)	0.44	2 (50)	2 (50)	0.60
• No	24 (89)	3 (11)		18 (67)	9 (33)	
Epithelial dysplasia						
• Mild	42 (100)	0 (0)	0.001	33 (79)	9 (21)	0.055
• Moderate	5 (83)	1 (17)		3 (50)	3 (50)	
• Severe	4 (57)	3 (43)		3 (43)	4 (57)	
Epithelial dysplasia						
• Low-grade	42 (100)	0 (0)	0.002	33 (79)	9 (21)	0.02
• High-grade	9 (69)	4 (31)		6 (46)	7 (54)	

**Table 2 jcm-08-01744-t002:** Evolution of the premalignant lesions in relation to histopathological diagnosis and SOX2 expression.

Variable	Number of Cases (%)	Progression to Carcinoma (%)	*P* *
Histopathological diagnosis			<0.001
• Low-grade dysplasia	42 (76)	2 (5)	
• High-grade dysplasia	13 (24)	10 (77)	
SOX2 > 10% positive nuclei			0.02
• Negative	51 (93)	9 (18)	
• Positive	4 (7)	3 (75)	
SOX2 any positive nuclei			0.01
• Negative	39 (71)	5 (13)	
• Positive	16 (29)	7 (44)	

* Fisher exact test.

**Table 3 jcm-08-01744-t003:** Univariate Cox cancer-free survival analysis in 55 patients with oral dysplasia categorized by dysplasia grading and SOX2 expression.

Variable	No.	Censored Patients (%)	Mean Cancer-Free Survival Time (95% CI)	*P*	Hazard Ratio	95% CI
Epithelial Dysplasia				< 0.001	19.08	4.09–89.01
• High-grade	13	3 (23)	100.69 (54.14–147.24)			
• Low-grade	42	40 (95)	181.59 (170.21–192.98)			
SOX2 > 10% positive nuclei				0.002	6.13	1.62–23.27
• Positive	4	1 (25)	69.00 (26.18–111.81)			
• Negative	51	42 (82)	191.80 (152.14–231.47)			
SOX2 any positive nuclei				0.002	5.75	1.68–19.74
• Positive	16	9 (56)	90.40 (64.14–116.66)			
• Negative	39	34 (87)	203.22 (162.30–244.15)			

*P*-values were estimated using the log-rank test.

**Table 4 jcm-08-01744-t004:** Multivariate Cox proportional hazards model to estimate oral cancer risk.

Variable	*P*	Hazard Ratio	95% CI
Histology (high-grade vs. low-grade)	< 0.0001	21.88	4.13–116.07
SOX2 > 10% (positive vs. negative)	0.196	3.0	0.57–15.89
SOX2 any (positive vs. negative)	0.021	5.83	1.31–26.01

**Table 5 jcm-08-01744-t005:** Univariate Cox cancer-free survival analysis in 55 patients with oral dysplasia categorized by both SOX2 and NANOG expression.

Variable	No.	Censored Patients (%)	Cancer-Free Survival Time (95% CI)	*P*	Hazard Ratio	95% CI
SOX2 > 10% positive nuclei and nuclear NANOG						
• Both negative	51	42 (84)	191.80 (152.13–231.46)	0.003	Ref	
• One positive	2	1 (50)	93.00 (32.01–153.98)		3.72	0.46–29.98
• Both positive	2	0 (0)	45.00 (8.00–97.92)		9.06	1.91–43.00
SOX2 > 10% positive nuclei and cytoplasmic NANOG						
• Both negative	45	38 (84)	171.10 (154.24–187.95)	< 0.0005	Ref	
• One positive	7	5 (71)	182.28 (101.53–263.03)		1.63	0.30–8.71
• Both positive	3	0 (0)	46.33 (15.66–76.99)		10.89	2.74–43.22
SOX2 any positive nuclei and cytoplasmic NANOG				< 0.0005		
• Both negative	36	31 (86)	175.49 (158.40–192.57)		Ref	
• One positive	13	11 (85)	215.99 (170.17–261.81)		1.091	0.20–5.85
• Both positive	6	1 (17)	44.50 (21.60–67.40)		11.36	3.18–40.60
SOX2 any positive nuclei and nuclear NANOG				< 0.0005		
• Both negative	39	34 (87)	203.22 (162.30–244.15)		Ref	
• One positive	14	9 (64)	97.76 (69.96–125.55)		4.62	1.23–17.29
• Both positive	2	0 (0)	45.00 (8.00–97.92)		14.82	2.69–81.56

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
