# Peer review of "SOX2 Expression Is an Independent Predictor of Oral Cancer Progression"

_jcm, 2019, doi:10.3390/jcm8101744_

Round 1
Reviewer 1 Report
In this manuscript, the authors show positive correlation for SOX2 expression for being an early predictor of oral cancer risk in patients with oral leukoplakia. Overall, this is an interesting study with a potential clinical application in near future.
Kindly find my comments:
Could the authors kindly summarize the patient demographics for the patients with oral epithelial dysplasia in a table similar to that of Supplementary table S1. Did the study identify any clear correlation in patients who are both tobacco /alcohol users and SOX2 expression?Author Response
Reviewer #1
Comments and Suggestions for Authors
In this manuscript, the authors show positive correlation for SOX2 expression for being an early predictor of oral cancer risk in patients with oral leukoplakia. Overall, this is an interesting study with a potential clinical application in near future.
Kindly find my comments:
Point 1: Could the authors kindly summarize the patient demographics for the patients with oral epithelial dysplasia in a table similar to that of Supplementary table S1.
Response 1: Following the reviewer’s suggestion, a new Supplementary Table S1 has been included in this revised version summarizing the most relevant clinical and pathological characteristics for the patients with oral epithelial dysplasia. Please note that previous Supplementary Tables S1 to S3 have been renumbered as Supplementary Tables S2 to S4 along the text. We also notice that the Supplementary Tables were mistakenly embedded in the previous version of the manuscript; however, all of them have been removed from the revised version, and are now provided as an independent file.
Point 2: Did the study identify any clear correlation in patients who are both tobacco /alcohol users and SOX2 expression?
Response 2: We thank the reviewer for raising this interesting question. Supplementary Table S3 (Supplem Table S2 in our previous version) already included data on the relationship of SOX2 expression with tobacco and alcohol consumption in our cohort of OSCC patients. Even though SOX2 expression was slightly more frequent in smokers and drinkers, these differences did not reach statistical significance (P = 0.63 and P = 0.39, respectively). Following the reviewer’s comment, we have also analyzed the possible combined effect of both tobacco and alcohol use. Among the 125 OSCC patients, 65 patients were both tobacco and alcohol users, and 29 (44.6%) of them showed positive SOX2 expression. However, a significant relationship was not observed (P = 0.32).
Regarding our series of 55 patients with oral epithelial dysplasia, information on tobacco and alcohol habits was only available for 31 patients, as now shown in new Supplementary Table S1. Among those patients both tobacco and alcohol users, only one patient showed positive SOX2 staining (>10% nuclei) and two cases SOX2 expression in any positive nuclei. None of these correlations were significant (P = 0.34 and P = 0.28, respectively).
Reviewer 2 Report
The manuscript of De Vicente and colleagues reports a retrospective study on the potential role of SOX2 transcription factor in early carcinogenesis stages of oral cancer. The authors evaluated SOX2 protein expression in samples of previously reported epithelial dysplasia (55 patients) and oral squamous cell carcinoma (125 patients) cohorts. The study suggests SOX2 expression as an independent predictor of oral cancer progression.
The manuscript is well written and findings clearly presented. Nevertheless, the previous report of the same group of authors on the same cohorts of patients is limiting the novelty impact of the manuscript. This and some other critical issues, detailed below, need to be solved in order to consider the manuscript for publication in the Journal of Clinical Medicine journal.
MAJOR POINTS
The patients with Oral Epithelial Dysplasia and those with OSCC included in this study were the same of a previously published report of the authors (https://www.mdpi.com/2077-0383/8/9/1376/htm). The methodologies of both manuscripts are almost the same, except for the specific marker analyzed, authors should refer to their previous report and include in this manuscript a summary of previously published methods/results together with the SOX2 novel findings. Moreover, in the same paper published by the authors (J. Clin. Med. 2019, 8(9), 1376; https://doi.org/10.3390/jcm8091376) they classified 42 (out of 55) patients as low-grade dysplasia and 13 as high-grade dysplasia according to the same WHO criteria cited in the present manuscript. If the criteria were the same, applied by the same authors to the same cohort of patients, how can the results different? A clarification on these points is needed. Paragraph 3.2 ("SOX2 Protein Expression in Oral Epithelial Dysplasia"). The representative pictures included in Figure 1 seem to indicate that the signal was zero (i.e. panel B) or very high (panel C). If this is the case, the choice of 10% as a cut-off of SOX2-positive cells would be completely clear. It would be interesting to identify also other thresholds: for example, among the positive samples, how many did show more than 50% positive cells? How many more than 75%? Row 206. What do the authors intend with "mainly located in the nucleus"? In the methods section, it has been reported that SOX2 positivity was evaluated and considered when nuclear staining was detected. If this is the case: 1) those cells showing cytoplasmic signal should have been ignored; 2) if SOX2-positive cells with cytoplasmic (non nuclear) signal have been identified, it should be reported and discussed in the manuscript.MINOR POINTS.
A) Caption of Figure 1, Row 171. As far as it can be seen from the pictures, magnification should not be 200x..
Author Response
Reviewer #2
Comments and Suggestions for Authors
The manuscript of De Vicente and colleagues reports a retrospective study on the potential role of SOX2 transcription factor in early carcinogenesis stages of oral cancer. The authors evaluated SOX2 protein expression in samples of previously reported epithelial dysplasia (55 patients) and oral squamous cell carcinoma (125 patients) cohorts. The study suggests SOX2 expression as an independent predictor of oral cancer progression.
The manuscript is well written and findings clearly presented. Nevertheless, the previous report of the same group of authors on the same cohorts of patients is limiting the novelty impact of the manuscript. This and some other critical issues, detailed below, need to be solved in order to consider the manuscript for publication in the Journal of Clinical Medicine journal.
Response: We thank the reviewer for the positive comments and meticulous revision. The reviewer is correct. SOX2 expression was certainly evaluated on the same patient cohorts and tissue samples (55 oral dysplasias and 125 OSCC) than our previous recently published study on NANOG (included as new Ref. 21: de Vicente JC et al. J. Clin. Med. 2019 Sept 3; 8 (9); https://doi.org/10.3390/jcm8091376). However, we truly think that this fact does not limit the novelty and relevance of our findings. On the contrary, we provide unprecedented evidence for the role of SOX2 expression as an early independent predictor of oral cancer risk in patients with oral leukoplakia. Altogether, both studies consistently and convincingly demonstrate a relevant role for these two pluripotency factors SOX2 and NANOG in early oral tumorigenesis rather than in tumor progression, and also underscore the potential clinical application for cancer risk-stratification.
Since both studies have been performed on the same subsets of patients and SOX2 and NANOG are functionally related proteins, we have taken advantage of these circumstances to assess the impact of combined expression of SOX2 and NANOG as predictor of oral cancer risk. These new data have been added as new Table 5. Interestingly, results consistently showed that patients harboring positive expression of SOX2 (either SOX2>10% or SOX2any) and NANOG (either cytoplasmic or nuclear expression) significantly exhibited a much higher risk of developing oral cancer, compared to patients with positive expression of either SOX2 or NANOG or those patients with negative expression.
MAJOR POINTS
Point 1: The patients with Oral Epithelial Dysplasia and those with OSCC included in this study were the same of a previously published report of the authors (https://www.mdpi.com/2077-0383/8/9/1376/htm). The methodologies of both manuscripts are almost the same, except for the specific marker analyzed, authors should refer to their previous report and include in this manuscript a summary of previously published methods/results together with the SOX2 novel findings. Moreover, in the same paper published by the authors (J. Clin. Med. 2019, 8(9), 1376; https://doi.org/10.3390/jcm8091376) they classified 42 (out of 55) patients as low-grade dysplasia and 13 as high-grade dysplasia according to the same WHO criteria cited in the present manuscript. If the criteria were the same, applied by the same authors to the same cohort of patients, how can the results different? A clarification on these points is needed.
Response 1: Our previous paper is now mentioned in this revised version (new ref 21) in Methods, Results and Discussion. Moreover, we are now providing valuable additional data showing the impact of combined expression of NANOG and SOX2 in oral dysplasias (new Table 5). Regarding the dysplasia grading, we must apologize for the confusion caused. The current WHO classification (now Ref 22) distinguishes three dysplasia grades i.e. low, moderate and severe dysplasia for oral epithelial lesions, as mentioned in the text (lines 161-162) and also shown in Table 1 and new Supplementary Table S1. In addition, a binary grading system has been proposed (Gale N, Blagus R, El-Mofty SK, Helliwell T, Prasad ML, Sandison A, Volavšek M, Wenig BM, Zidar N, Cardesa A. Evaluation of a new grading system for laryngeal squamous intraepithelial lesions--a proposed unified classification. Histopathology 2014;65:456-64) to classify oral dysplasias similar to the WHO classification (4th edition) currently accepted for laryngeal dysplasias into the categories: low-grade and high-grade (including moderate and severe dysplasias) (WHO Classification of Head and Neck Tumours. WHO/IARC Classification of Tumours, 4th ed.; El-Naggar, A.K., Chan, J.K.C., Grandis, J.R., Takata, T., Slootweg, P.J., Eds.; IARC: Lyon, France, 2017; Volume 9, ISBN 978-92-832-2438-9). Nevertheless, this binary system has not yet been validated for use in the oral cavity. Therefore, according to the WHO 2017 classification there is not a unique criteria or grading system for oral epithelial dysplasias, and as a consequence diagnostic reproducibility is still limited.
In this respect, the apparent discrepancy between our previous paper and the present study lies on the fact that in NANOG paper the binary WHO classification (low vs high-grade) was used while in the present study the WHO three-tier grading of oral dysplasia (as shown in Table 1). However, for statistical purposes, in the subsequent analyses performed (shown in Tables 2-4) this 3-tier grading was dichotomized into the categories: mild-moderate vs severe dysplasia. Nevertheless, to facilitate comparisons between the results from both studies and to avoid confusion, Table 1 also includes the very same binary WHO classification (low-grade vs high-grade dysplasia) used in our previous publication. Similarly, this binary grading have now been used in all the subsequent analyses, and the text and Tables 2-4 have accordingly been amended. Figure 2 now shows Kaplan-Meier curves for both the WHO three-tier classification and the binary dysplasia grading. Please note that previous Figure 2D is now Figure 3, and previous Figure 3 is now Figure 4.
Noteworthy, the main findings of our study still remain the same by using the binary dysplasia grading (low-grade vs high-grade). Thus, SOX2 expression was found a significant independent predictor of cancer risk in the multivariate analysis, which reflects the robustness of SOX2 as an independent risk marker. However, this binary histological grading showed superior predictability than SOX2 expression in the studied patient cohort. According to this observation, we cannot conclude now that SOX2 expression predicts oral cancer risk “beyond histological grading”. Hence, this has been amended in the title and text.
Point 2: Paragraph 3.2 ("SOX2 Protein Expression in Oral Epithelial Dysplasia"). The representative pictures included in Figure 1 seem to indicate that the signal was zero (i.e. panel B) or very high (panel C). If this is the case, the choice of 10% as a cut-off of SOX2-positive cells would be completely clear. It would be interesting to identify also other thresholds: for example, among the positive samples, how many did show more than 50% positive cells? How many more than 75%?
Response 2: We must clarify that in addition to the cut-off of SOX2-positive >10% nuclei, SOX2 staining in the dysplastic areas was also scored considering any positive nuclei, as specified in Methods (lines 142-145). Since CSC-like subpopulations are usually limited to a very small percentage of cells, any SOX2-positive nuclei could be meaningful. In fact, SOX2any showed superior predictive value compared to SOX2>10. Also, note that only 4 cases were detected positive for SOX2>10, and therefore results will not be improved by using 50% or beyond as additional cut-off points.
Point 3: Row 206. What do the authors intend with "mainly located in the nucleus"? In the methods section, it has been reported that SOX2 positivity was evaluated and considered when nuclear staining was detected. If this is the case: 1) those cells showing cytoplasmic signal should have been ignored; 2) if SOX2-positive cells with cytoplasmic (non nuclear) signal have been identified, it should be reported and discussed in the manuscript.
Response 3: Indeed, positive SOX2 expression was detected in the nucleus of tumor cells, while expression was negligible in stromal cells and normal epithelia. To avoid confusion, this has now been corrected (line 226).
MINOR POINTS
Point 4: A) Caption of Figure 1, Row 171. As far as it can be seen from the pictures, magnification should not be 200x.
Response 4: Magnification was 20x. Thanks for noticing this error, which has now been amended in legend for Figure 1.
Round 2
Reviewer 2 Report
The authors modified the manuscript taking into account the concern expressed by this reviewer and solved some of the issues raised in the first round of review.
With reference to the partial overlapping of this work with the one recently published by the same group, the reviewer acknowledges the addition of combined analyses of NANOG together with SOX2, which is now adding some of the previously lacking novelty to the manuscript, with particular regards to paragraph 3.2 of the revised manuscript.
Nevertheless, there is another finding, claimed by the authors in this manuscript, that has been already reported in their previous paper. In the text of paragraph 3.5 (row 247) authors report 22 cases (12,7% of 172 included in the TCGA cohort) harboring SOX2 mRNA up-regulation, but in Figure 4A, the panel reports 22% in the corresponding oncoprint representation. The correct percentage is the one reported in the text, as already published by the authors in their previous work (now ref. 21). The reviewer would suggest to correct the figure and to consider, again, to add the citation to their paper (now ref. 21), since this analysis has been already reported there and cannot be considered as new.
Another important issue related to results included in paragraph 3.5 is the following. How do the authors discuss their findings of protein expression of SOX2 in their OSCC cohort and the findings of their report about RNAseq and CNV in the cohort from TCGA? In this context, the discussion could not be limited, as in the present version of the manuscript (rows 319-325) to analyze possible mechanisms underlying discrepancies between SOX2 gene amplification and SOX2 mRNA expression. It is expected some discussion, eventually with bibliographically supported speculations, on how these data correlate with the protein expression detected by the authors in their cohort of OSCC patients.
Minor point: Row 205, reference should be 21 and not 23.
Author Response
Comments and Suggestions for Authors
The authors modified the manuscript taking into account the concern expressed by this reviewer and solved some of the issues raised in the first round of review.
With reference to the partial overlapping of this work with the one recently published by the same group, the reviewer acknowledges the addition of combined analyses of NANOG together with SOX2, which is now adding some of the previously lacking novelty to the manuscript, with particular regards to paragraph 3.2 of the revised manuscript.
Response: We thank the reviewer for considering that the previous concerns were adequately addressed, and for highlighting the added value and novelty of the new data included in the revised version of the manuscript.
Point 1: Nevertheless, there is another finding, claimed by the authors in this manuscript, that has been already reported in their previous paper. In the text of paragraph 3.5 (row 247) authors report 22 cases (12,7% of 172 included in the TCGA cohort) harboring SOX2 mRNA up-regulation, but in Figure 4A, the panel reports 22% in the corresponding oncoprint representation. The correct percentage is the one reported in the text, as already published by the authors in their previous work (now ref. 21). The reviewer would suggest to correct the figure and to consider, again, to add the citation to their paper (now ref. 21), since this analysis has been already reported there and cannot be considered as new.
Response 1: We must clarify that the oncoprint shown in Figure 4A actually represents all the different SOX2 alterations present in the subset of 172 OSCC from the TCGA. Thus, the oncoprint representation depicts SOX2 mRNA upregulation together with SOX2 gene amplification and missense mutations reaching a total percentage of 22%, while specifically SOX2 mRNA upregulation was detected in 22 cases (13%) as mentioned in the text. Please note that the oncoprint shown in Figure 4A is different to that previously published (in ref. 21). Accordingly, the oncoprint included in NANOG paper was aimed at comparing mRNA expression of NANOG with other CSC-related factors such as SOX2, OCT4 and PDPN, whereas the oncoprint in the present study (Fig 4A) was intended to investigate possible underlying mechanisms of altered expression of SOX2 in OSCC. To this purpose, SOX2 mRNA upregulation and SOX2 gene amplification were assessed and correlated using data from the TCGA.
To avoid confusion the text has been modified, a new sentence has been added (lines 220-221) and also the ref. 21 has now been included (line 222), now it reads: “As shown in Figure 4A, SOX2 gene alterations were present in a total of 38 (22%) of 172 OSCC patients. 22 (13%) cases harbored SOX2 mRNA up-regulation as previously reported [21], and 23 (13%) cases harbored SOX2 gene amplification.”
Point 2: Another important issue related to results included in paragraph 3.5 is the following. How do the authors discuss their findings of protein expression of SOX2 in their OSCC cohort and the findings of their report about RNAseq and CNV in the cohort from TCGA? In this context, the discussion could not be limited, as in the present version of the manuscript (rows 319-325) to analyze possible mechanisms underlying discrepancies between SOX2 gene amplification and SOX2 mRNA expression. It is expected some discussion, eventually with bibliographically supported speculations, on how these data correlate with the protein expression detected by the authors in their cohort of OSCC patients.
Response 2: Following the reviewer’s recommendation, we have further discussed the findings on SOX2 protein in our OSCC cohort together with the results from the analyses performed using the TCGA cohort of 172 OSCC patients (lines 298-302). References have been modified accordingly.
Point 3: Minor point: Row 205, reference should be 21 and not 23.
Response 3: Thanks for noticing this error, which has now been corrected (line 185 in this new version).